# Fitting the BumpHunter test statistic distribution and global p-value estimation

Louis Vaslin[1], Samuel Calvet[1], Vincent Barra[2], and Julien Donini[1]

[1]LPC, Université Clermont Auvergne, CNRS/IN2P3,
Clermont-Ferrand; France
[2]Université Clermont Auvergne, CNRS, LIMOS, UMR 6158,
Clermont-Ferrand; France

November 14, 2022

## Abstract

In high Energy Physics, it is common to look for a localized deviation in data with respect to a given reference. For this task, the well known BumpHunter algorithm allows for a model-independent deviation search with the advantage of estimating a global p-value to account for the Look Elsewhere Effect. However, this method relies on the generation and scan of thousands of pseudo-data histograms sampled from the reference background. Thus, accurately calculating a global significance of $5\sigma$ requires a lot of computing resources. In order to speed this process and improve the algorithm, we propose in this paper a solution to estimate the global p-value using a more reasonable number of pseudo-data histograms. This method uses a functional form inspired by similar statistical problems to fit the test statistic distribution. We have found that this alternative method allows to evaluate the global significance with a precision about 5% up to the $5\sigma$ discovery threshold.

## Introduction

BumpHunter [1] is a well known algorithm that searches for a localized deviation in a data distribution with respect to a reference background and assesses its local and global statistical significance. Such an algorithm is useful in various domains, including in the search for new resonances in High Energy Physics (HEP) [2, 3]. In order to provide a public and practical implementation of the BumpHunter algorithm, a Python version named pyBumpHunter has been recently developed [4]. This implementation provides several extensions to the original algorithm proposed in [1], including signal-injection sensitivity test, 2D BumpHunter, side-band normalization and multi-channel combinations.

The features proposed in pyBumpHunter can be applied to perform a model independent signal search. However, as discussed in [5], the way the global-value is computed could be improved. Indeed, BumpHunter relies on the sampling of a large number of background-only distributions to compute the global p-value. With such a method, the number of pseudo-data distributions required to estimate a global significance around $5\sigma$ becomes very high (several millions), which makes it quite time consuming.

In order to solve this issue, we propose a solution to calculate high global significances at a lower cost using a fit of the BumpHunter test statistics distribution. This procedure has been studied in order to determine any potential bias on the global significance. The code used to produce the results presented in this paper is available on GitHub [6].

## 1 The BumpHunter algorithm

The BumpHunter algorithm searches for a deviation in an observed data histogram with respect to a reference model by scanning the distribution with several variable-

width windows. Local p-value are calculated for all tested interval positions (and widths), and the interval with the smallest p-value is retained as the one which presents the most significant deviation with respect to the reference. The selected local p-value $p$ is transformed into a test statistic value $t$ with:

$$t = -\ln(p). \tag{1}$$

The procedure is repeated on thousands of pseudo-data histograms generated by smearing the bin content of the reference with a Poisson law. This gives a distribution of local p-values that corresponds to the minimum p-values that were selected for each pseudo-data histogram. Following equation 1, this distribution is transformed into a test statistic distribution that will be used to compute the global p-value and significance. This procedure is used to account for the Look Elsewhere Effect [7].

The problem arises when the test statistic value associated with observed data is higher than all the test statistic values associated with the background-only pseudo-data. In this case the global p-value will be exactly 0, which corresponds to an infinite global significance. In addition, the smallest non-null global p-value that can be computed with this method is equal to the inverse of the total number of generated pseudo-data. Therefore in order to reach $5\sigma$ global significance with a sufficient number of pseudo-data histograms that have a test statistic value greater than the one observed in data, it is necessary to produce and scan millions of pseudo-data histograms.

## 2 Fitting the BumpHunter test statistic distribution

In order to reduce the number of pseudo-data distributions required to compute the global p-value, one solution is to fit the test statistic distribution with a function. This function can then be used to extrapolate the distribution to higher values, allowing to reach smaller p-values (and higher significances). A potentially suitable function can be found using the "p-value hacking" method proposed in [8].

In this work, we consider the following problem: a test is performed $m$ times on a population of $n$ individuals. For each of these tests, a p-value is assessed, resulting in a distribution of $m$ p-values. Then, in order to retain an unique p-value for this population only the minimum p-value among these $m$ tests is considered. This procedure can be repeated many times on different populations, each of them being assigned a minimum p-value. In the end, a distribution of minimum p-values is obtained. The analytic form of this distribution $\varphi$ is given by [8]:

$$\varphi(p; p_M) = m \, e^{\text{erfc}^{-1}(2p_M)\left(2\text{erfc}^{-1}(2p) - \text{erfc}^{-1}(2p_M)\right)}$$
$$\left(1 - \frac{1}{2}\text{erfc}\left(\text{erfc}^{-1}(2p) - \text{erfc}^{-1}(2p_M)\right)\right)^{m-1} \tag{2}$$

with $p$ the minimum p-value, $m$ the number of tests, $p_M$ the true median of the p-value distribution and erfc the complementary error function. Here the denomination "true median" comes from the fact that, in such a procedure, the p-value associated with each population is biased since the most favorable occurrence is chosen in a set of repeated tests. Thus, this effect has to be accounted for when computing the median of the distribution.

One can relate the local p-value distribution obtained with BumpHunter with the problem treated in [8]. In the case of BumpHunter, the $n$ individuals of the population can be identified to the number of events in the tested intervals. As for the $m$ tests from which the minimum local p-value is chosen, they correspond to all the different intervals that were tested for a given pseudo-data histogram. Then, the test is repeated on different populations, or in the case of BumpHunter, on different pseudo-data histograms. With this equivalence established, it is reasonable to suppose that the formula given in equation 2 could be used to fit the BumpHunter (minimum) local p-value distribution. Equation 1 can then be used to transform the minimum p-value distribution $\varphi(p)$ into the BumpHunter test statistic distribution $\Phi(t)$.

Assuming this procedure would give a suitable analytical function to fit the BumpHunter test statistics distributions, there are two aspects that should be evaluated.

The first one concerns the number of individuals $n$ that compose the tested population. In the development presented in [8], this number is assumed to be high enough to allow the use of the Central Limit Theorem, so that equation 2 could be derived. In the case of BumpHunter, the relevant number corresponds to the number of individuals that fall in the tested intervals. As this number of events depends on the available statistics as well as on the shape of the scanned distribution, there are cases where the limit condition is not verified, for example when considering an exponentially falling background with low statistics in the tail.

Another potential limitation comes from the $m$ tests from which the minimum p-value is chosen. In the problem treated in [8], the author states from the beginning that the $m$ tests are independent. In the case of BumpHunter, this condition is usually not verified. The only case where the condition stands is when the width of the scan window is equal to one bin. In any other cases, the tested intervals will overlap such as to induce correlations between the different tests, making them not fully independent. For these two reasons, studies are required in order to verify if the solution proposed in [8] applies correctly to the case of BumpHunter or if any potential biases should be taken into account. These studies are the object of the next section.

## 3   Test of the fit procedure and bias evaluation

In order to test the behaviour of the BumpHunter test statistic fit, the following procedure has been established :

- A data histogram and a reference histogram are generated by sampling from the same probability density function.

- A low-statistics test statistic distribution is produced using $5 \times 10^4$ pseudo-data histograms. The global p-value $p_{fit}$ is evaluated by, first, fitting the test statistic distribution using equation 2 and then integrating this function for different thresholds of test statistics values ($t_{data}$).

- A high-statistics test statistic distribution is produced using, this time, $2 \times 10^7$ pseudo-data histograms. The higher statistics allows to compute a global p-value $p_{BH}$ directly from the $t$ distribution without relying on a fit. Again, this p-value will be evaluated for different thresholds of data test statistics values.

- A bias factor ($R$) on the global significance is defined as the ratio between the global significance calculated using the high-statistics test distribution ($\sigma_{BH}$) and the one calculated using the low-statistics distribution fit function ($\sigma_{fit}$) :

$$R = \frac{\sigma_{BH}}{\sigma_{fit}} \tag{3}$$

In order to cover multiple cases, this procedure is applied with different background shapes and statistics. For the first test, we use an uniformly distributed background with $10^5$ events in the range $[0, 35]$ and 20 bins of equal width. This example covers the case where the available statistics in each bin is high enough ($n > 30$). The width of the scan window is set to 1 bin to ensure that tested intervals don't overlap with each other, making all intervals independent. A second test is performed in the same setup, but using an exponentially distributed background with 1k events generated in the range $[0, 35]$ with the same binning as previously (the exponential has a scale of 3.5). Thus, the case where some intervals have a low statistic, in the tail of the distribution, can be studied. Finally, a third test is performed using the same uniform distribution as in the first test, but with a scan window width set to 5 bins. This way, there are up to 4 overlapping bins between intervals and this example serves to study a case where the $m$ tests from which the minimum p-value is selected are not fully independent.

Figure 1 shows the results obtained in the first, say optimal, case. The left panel shows the test statistics distribution obtained from the $5 \times 10^4$ pseudo-experiment (blue histogram) together with the fitted function (orange line). The fit is of good quality, with a $\chi^2/ndf$ test value of 1.45. The panel on the right shows the evolution of the global significance, as function of $t_{data}$, when it is calculated either with the fit

function (solid orange) or by using the direct high-statistics method (dashed blue). The two curves are very close and can not be distinguished by eye. The bias ratio as a function of $t_{data}$ confirms that the global significances are compatible within 1%. The variations that can be observed when $t_{data} > 15$ are due to the low statistic available in the tail of the test statistic distribution used to compute the global p-value using the direct method.

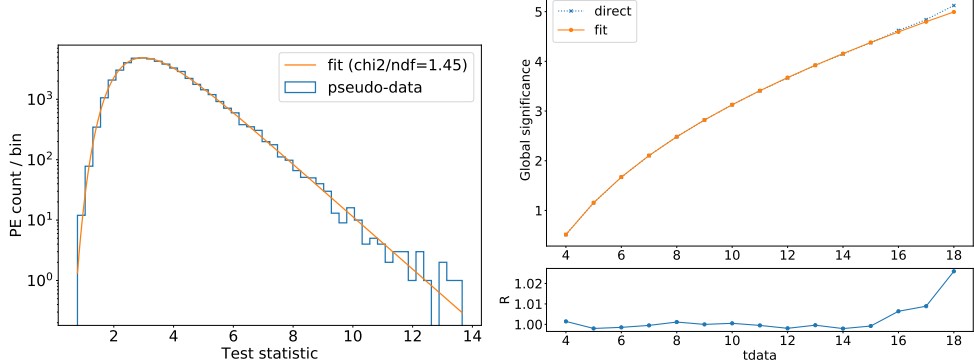

Figure 1: Results of the BumpHunter test statistic fit in the ideal case. The left plot shows the test statistic histogram associated to pseudo-data with the fitted function. The top plots show the evolution of the global significance obtained with the direct method and using the fit as function of $t_{data}$ together with the bias ratio.

Figure 2 shows the results obtained when the statistics is low in the tail of the scanned histograms. The quality of the fit shown on the left panel is quite poor, indeed, the test statistic distribution is not smooth anymore and presents numerous spikes. In fact, when the available statistic in the selected interval is low, there are very few possible values that the local p-value can take compared to the first case. Nonetheless, the right panel shows that the evolution of the global significance obtained using the fit function closely follows the one obtained with the direct method. The ratio confirms that the bias induced by the fit on the global significance is within 4% for all the tested values of $t_{data}$. Only the last points corresponding to $t_{data}$ greater than 16 diverge because of the available statistics for the direct method.

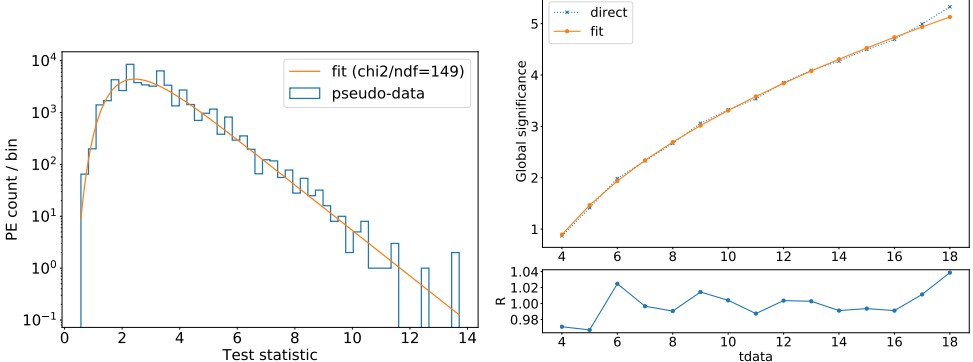

Figure 2: Results of the BumpHunter test statistic fit in the case when the statistic is low in the tail of the scanned distributions. The left plot shows the test statistic histogram associated to pseudo-data with the fitted function. The right plot shows the evolution of the global significance obtained with the direct method and using the fit as function of $t_{data}$ together with the bias ratio.

Figure 3 illustrates the case when the tested intervals overlap. The left panel shows that the fit is of good quality, with a $\chi^2/ndf$ value of 1.06, and the test statistic distribution is as smooth as in the first case. However, the right panel shows that the global significance obtained using the fit function diverge from the ones obtained with the direct method when $t_{data}$ is greater than 9. This tendency is confirmed when computing the bias ratio. This bias increases with $t_{data}$ and reaches up to 2%

for $t_{data} = 13$. For higher values, the global significance computed using the direct method shows significant variations due the reasons discussed in the previous cases.

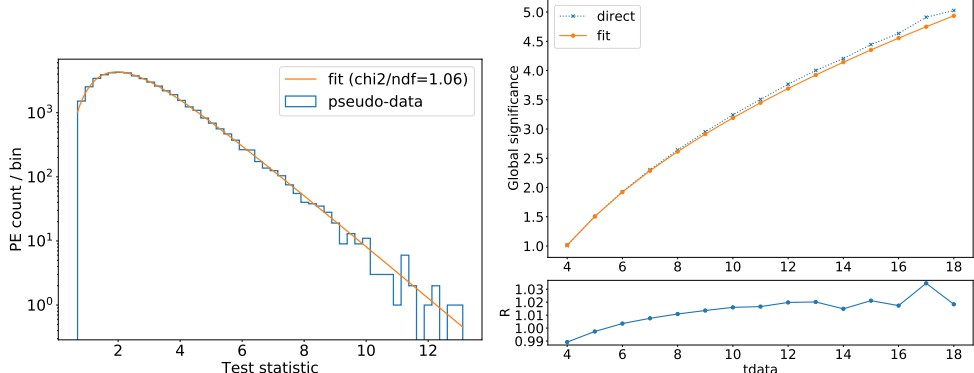

Figure 3: Results of the BumpHunter test statistic fit in the case when the tested intervals overlap. The left plot shows the test statistic histogram associated to pseudo-data with the fitted function. The right plot shows the evolution of the global significance obtained with the direct method and using the fit as function of $t_{data}$ together with the bias ratio.

In addition to the three cases presented here, other tests have been performed using different scan window widths, as well as reference background distributions of different statistics and binning. The results have shown that the bias induced by the fit on the global significance stays below 5% in any case. We have also seen that the shape of the background distribution doesn't affect much the shape of the test statistic distribution as long as the statistic available in the scanned distribution is high enough.

# Conclusion

We have defined a procedure that allows to compute the global p-value by fitting the BumpHunter test statistics distribution. This procedure is relevant when the global significance becomes too high to be computed efficiently using the direct method originally defined in [1]. We could verify that, when all the validity conditions of the fit function are met, there is no bias on the global significance induced by the fit procedure. We could also verify that, in the cases where these conditions are not met, the bias on the global significance doesn't go over 5% up to significance of $5\sigma$. Other tests using alternative background distribution with different binning and multiple scan window widths tend to confirm this statement. This encouraging result motivates the implementation of an automatic fit procedure in the future releases of pyBumpHunter.

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
