# Peer review of "Fitting the BumpHunter test statistic distribution and global p-value estimation"

_SciPost Physics Codebases_

## Round 1 · Referee Report · Anonymous (Referee 1) · 2023-12-9

Strengths

1- the paper is cleanly written and only requires very minor vetting to be publishable

2- the topic is of interest for the domain of high-energy physics and astrophysics

3- the paper describes a useful approximation to reduce CPU cost of significance calculations

Weaknesses

1- a more thorough comparison of the test statistic under different experimental conditions would be interesting

Report

The paper discusses an approximated calculation of the global p-value of bump searches in histograms. The description is clear and results discussed are convincing. I would have been happy with a more systematic study of the cases when the method breaks down (higher significances, different PDFs, non-Poisson cases) but I believe the paper can be published even without those studies.

Requested changes

1- it would be better to avoid apostrophes (don't -> do not, etc.) in written English. I found three instances in the text

2- Abstract: "a lot of" -> suggest "significant"; "speed this process" -> "speed up this process"

3- Introduction, 2nd paragraph: "global-value" -> "global p-value".

4- Introduction, last paragraph: the two sentences "In this case the global p-value ... ... In addition, the smallest..." are redundant, in the sense that the second alone would suffice. The first is questionable as one would likely not estimate p=0 upon seeing no observations above the cutoff, but rather revert to other methods anyway.

5- Sec. 2 it would be good to write down explicitly the erfc function for the benefit of the readers.

6- Sec. 2 Page 3 first paragraph: "number of individuals that fall ... As this number of events" -> not sure why we start off with individuals and then mix up with events. Better stick to events everywhere, to avoid confusions.

7- Fig. 2 I suppose the chi2/ndf quoted there misses a comma (149? -> 1.49)

8- The paper never mentions the speed up factor in using the approximation, it would be good to have it in the abstract and/or conclusions to give more value to the method.

---

## Round 1 · Referee Report · Anonymous (Referee 2) · 2023-12-30

Strengths

- interesting application for estimating statistical global significances in high energy physics discovery
- novelty
-relatively easy way of implementing

Weaknesses

-make general statements out of a limited number of examples
-not totally correct statistical description

Report

The draft “Fitting the BumpHunter test statistic distribution and global p-value estimation” presents an interesting method for evaluating global p-values in cases where normal “toy-MC” sampling is not achievable. It presents a novel and practical implementation for real cases, although it is based on a statistical model only valid under several assumptions that are not usually fulfilled and the effect on the results is not sufficiently explained.
I believe it is worth publishing but some of these have to be solved.

Requested changes

Following are listed some comments in sequential order:
Section 1
even if not central for the paper should be revised, many things are unclear, and the statistical language is not 100% correct. There is some confusion between the p-value, as a quantity we want to estimate, and the estimation made from bumphunter. In fact, how this is estimated is not explained. Is it just a poisson upper tail? Does the package provide several alternatives? Note that eq (1) does not “transform” p into a test statistic, just uses a functional transformation of a test statistic into a form that is usually better behaved. “Smearing by a Poisson law”, do you mean bootstrapping from the data distribution assuming poisson law? The last paragraph is not fully correct, your estimation of the p-value will be zero, but certainly not the p-value, you could not use point estimate in this case.
Section 2
You need to more clearly state the conditions assumed in [8] to obtain (2), they are mentioned in several places, but not too clearly. Is just n->infinity and independent m?
“the n individuals of the population can be identified to the number of events in the tested intervals”, Is that right? I would have assumed the relevant variable would be the number of bins or the number of fits. Did you check the performance in cases with few bins?

“With this equivalence established, it is reasonable to suppose that the formula given in equation 2..” How do we know it is reasonable? There is no clear connection of this statement with what is mentioned above. Similar comment to the sentence “assuming this procedure would give…” it is a circular argument, if we assume that it is suitable, then it will be.

As for the quoted concerns, you have explored the dependence with n, although it would be nice to see a more systematic study. It would be interesting to see how the performance changes with respect to the width. As said above, it would be interesting to see checks on the number of bins.

Section 3

Was the number of pseudo-experiments varied? Why choose exactly 5E4, could we use fewer or get better results with more?

The bias factor as a ratio of significance is not too informative, a 10% on a five sigma significance implies an order of magnitude in the p-value. Why not using ratios of p-values? In fact, plotting the cumulative upper tail would be much more informative.

In several places you quote “good fits” based on chi^2/ndof, note this is not a good quantity to measure the goodness of fit and should instead use a p-value. In fact, if I correctly interpret your numbers, for example fig 1 you might have more than 100 bins, chi^2=145 for ndof=100 gives a pvalue of 0.2% which is certainly not too good.

How can you be sure that the departures in fig 1 close to five sigma is due to the low stats? Note that region is the relevant one if you want to use this for discovery. In fact, it is the justification you use in the introduction for the interest of the method. Did you try to estimate the uncertainty (at least statistically) in your prediction?

How extreme is the scenario in fig 2, is it realistic or still conservative? It is important to know when the assumptions in [8] break.

Fig 3 is interesting, because tests the other limitation. A more detailed study would be interesting. Same questions as above is this example a worst-case scenario or a conservative one? Where is the limit?

It would be interesting to see a numeric summary of all these additional tests and more.

Although I agree that the results are encouraging, I don’t think you can conclude from the tests shown that the bias does not go beyond 5% in “all” cases. What if you are running a background free analysis and finding a signal with a handful of events? What if you are looking for a very wide signal and the overlap is large?

---

## Editorial Decision

awaiting_resubmission